# Fatigue Tests and Failure Mechanism of Rib-to-Deck Welded Joints in Steel Bridge

Chuanbin Fan [1,2], Letian Da [3,*], Kangchen Wang [1,2], Shenyou Song [1,2] and Huanyong Chen [1,2]

1 Guangdong Highway Construction Co., Ltd., Guangzhou 510000, China
2 Shenzhen-Zhongshan Link Administration Center, Zhongshan 511462, China
3 Department of Bridge Engineering, Southwest Jiaotong University, Chengdu 610031, China
* Correspondence: letianda@my.swjtu.edu.cn

**Abstract:** The fatigue cracks of rib-to-deck welded joints in an orthotropic steel deck are one of the critical conundrums that restrict the sustainable development of steel bridges. Double-sided welded joints were achieved by introducing internal welding technology to overcome the initial "crack-like" manufacturing defects at the welding roots of single-sided welded joints. Through the observation of the macro-section of the welded joint, the differences between single-sided welding, partial-penetration welding, and full-penetration welding in rib-to-deck welded joints are compared. Relying on the Shenzhen–Zhongshan link, the fatigue failure and mechanism of single-sided rib-to-deck welded joints are clarified by the nominal stress and structural stress methods, and those of double-sided welded joints were determined through fatigue tests. The fatigue strength of rib-to-deck welded joints is higher than the FAT90 of Chinese standard and the FAT C joints of American standard. The fatigue strength of double-sided rib-to-deck welded joints is significantly higher than that of single-sided welded joints. The fatigue strength of rib-to-deck welded joints is within a $\pm 2\sigma$ range of main *S-N* curves by the structural stress method. It is suggested to adopt the double-sided welded joints in practice, and to ensure that the penetration rate is beyond 80%. The trial data are limited, and further tests are needed to confirm the results.

**Keywords:** steel bridge; fatigue damage mechanism; rib-to-deck welded joint; fatigue tests; fatigue strength

## 1. Introduction

Orthotropic steel decks have the distinct advantages of high strength, light weight, high load-carrying capacity, and broad applicability. They are increasingly used in bridge engineering [1–4]. Due to multiple factors, such as the service environment, loading conditions, and welding manufacturing defects, fatigue problems are prominent in orthotropic steel decks. In recent years, there have been more and more fatigue cracks occurring in in-service steel bridges [5–8]. As shown in Figure 1, several fatigue cracks were found in the Humen Bridge within the past 10 years. Mairone et al. evaluated the fatigue damage of a highway viaduct by nominal stress and showed that its welded joints could no longer meet the fatigue performance requirements [9]. Moreover, once there is a fatigue crack, it will develop quickly, and it is difficult to monitor the service performance of the bridge. Thus, fatigue has become a bottleneck problem that inhibits the development and application of steel bridges and has aroused widespread concerns in the engineering community.

The leading causes of the fatigue cracking of rib-to-deck welded joints include insufficient welding penetration, slagging in grooves or at welding roots, assembly errors, and the improper handling of welded joints. Scholars have studied the effects of variable rib thicknesses, deck thicknesses, and the penetration rate of weld seams on the fatigue performance of rib-to-deck welded joints. Dung et al. compared the fatigue properties of single-sided rib-to-deck welded joints with different penetration rates through fatigue

tests and concluded that a 100% penetration rate was more conducive to improving the fatigue properties of single-sided rib-to-deck welded joints [10]. Sim et al. showed a different pattern. A shallower welded penetration (80% welded penetration) also appeared to have a slightly higher fatigue resistance than a deeper one (100% welded penetration) [11]. Li et al. studied the influence of permeability on the fatigue properties of single-sided rib-to-deck welded joints using the structural stress method, and the results showed that welded penetration had different effects on different fatigue failure modes [12]. Zheng et al. improved the fatigue performance of single-sided rib-to-deck welded joints by changing the thickness of U-ribs [13]. Through finite element analysis, Ya et al. found that in a large-rib-deck model, increasing the thickness of the deck and improving the rigidity of the pavement were conducive to improving the life of the orthotropic steel deck [14]. Kolstein obtained the fatigue strength of rib-to-deck welded joints under different fatigue failure modes by summarizing a fatigue test 3. Cheng et al. observed and evaluated the fatigue crack initiation and propagation process, fatigue failure mode, characteristic fatigue life, and degradation of vertical rigidity by fatigue tests [15]. Liu et al. found through fracture mechanics analysis that increasing the thickness of the deck and reducing the initial crack surface length and depth can improve fatigue performance [16].

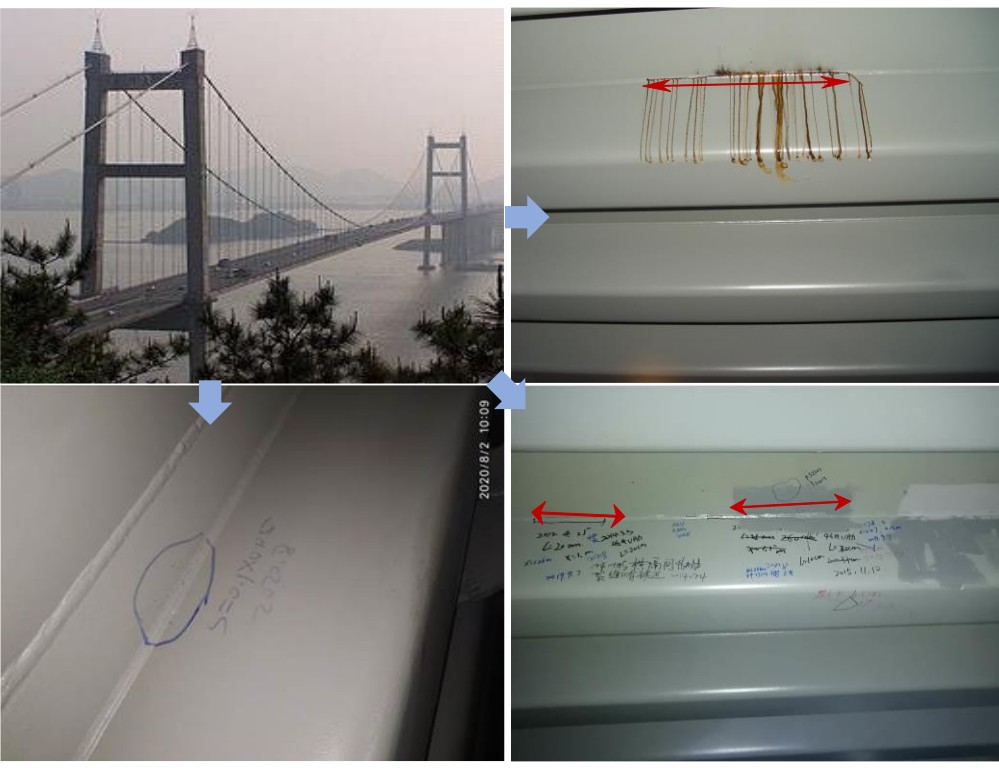

**Figure 1.** Typical fatigue cracks of Humen bridge.

Scholars have tried to improve the fatigue performance of rib-to-deck welded joints through methods such as grinding, ultrasonic impact, and shot-peening, which improve the shapes of weld seams and reduce residual welding stresses. Yamada et al. studied the effect of ICR (Impact Crack-Closure Retrofit) treatment on fatigue cracks, which reduced their growth rate [17]. Kinoshita et al. investigated the effects of fatigue strength improvement of shot-peening on welded joints [18]. Tai et al. studied the improvement of welded joints by peening treatment from the viewpoints of stress concentration, residual stress distribution, plastic deformation, and crack propagation behavior [19]. They have also proposed combining UHPC (ultra-high-performance concrete) with steel deck plates to reduce the stress of welded joints, thus increasing the service life of steel deck plates [20–22].

With the process of automated welding technology, China has developed an automated U-rib internal welding technology with independent intellectual property rights and proposed a double-sided welding method for rib-to-deck welded joints. This technology has now been widely used in such major construction projects as the Zhuankou Yangtze River Bridge, the Zhaoqing Xijiang Bridge, and the Shenzhen–Zhongshan Link [23–26]. Da et al. compared the difference in stress response between single-sided rib-to-deck welded joints and double-sided rib-to-deck welded joints under vehicle loads through finite element analysis. The results showed that the single-sided rib-to-deck welded joints had greater stress concentration at the welding roots [23]. Cui et al. studied the fatigue properties of double-sided rib-to-deck welded joints with small-sized specimens and showed that they had high fatigue strength [24]. Yang et al. evaluated the fatigue properties of single-sided rib-to-deck welded joints and double-sided rib-to-deck welded joints with the structural stress method [25]. Zhu et al. obtained the stress response of double-sided rib-to-deck welded joints under vehicle loads through real bridge tests [26].

This paper takes single-sided rib-to-deck welded joints and double-side rib-to-deck welded joints its the research objects. In the introductory section, the relevant research content and progress are expounded. Then, the welding requirements of rib-to-deck welded joints are clarified. Through observation of the macro-section, the differences between the rib-to-deck welded joints in their weld shape, unfused zone, and other aspects are compared (Section 2). In order to ensure that the test model is consistent with a real bridge in terms of weld quality and stress characteristics, nine full-size specimens are designed (Section 3). The fatigue properties of rib-to-deck welded joints were obtained by various loading methods. The fatigue properties of single-sided rib-to-deck welded joints and double-sided rib-to-deck welded joints were compared by the nominal stress method and the structural stress method (Section 4). Finally, all the research contents are summarized in Section 5.

## 2. Shape Comparison of Rib-to-Deck Welded Joints

Welding should be carried out indoors or in wind-proof and rain-proof facilities, with no more than 80% environmental humidity. When welding low-alloy steel, the ambient temperature should not be lower than 5 °C. To weld low-carbon steel, the ambient temperature should be no lower than 0 °C. If the environmental temperature or humidity cannot meet these requirements, necessary technological measures should be implemented before welding.

Before the welding, harmful substances in the working area must be completely cleared out. All the equipment used should be examined to ensure that the working conditions are normal and the instruments are functional, complete, and reliable. After the welding, slags on the surface of any weld seam and welding spatter on both sides of the seam should be cleared out.

### 2.1. Rib-to-Deck Single-Sided Welded Joint

The penetration rate of weld seams should reach 80% or above on rib-to-deck single-sided welded joints. Apply a flux-cored $CO_2$ shielded groove welding method to weld the rib-to-deck joints, with groove angles ranging from 50° to 55°. Use the two-pass welding forming method and ship-position welding on the weld seams. During the test process, adjust the welding current, voltage, and speed and the welding gun's angle, swing amplitude, and swing frequency. After welding, first check the welding appearance. On the weld seams have a good appearance, conduct a cross-section acid-etching test to check the unfused depth, as shown in Figure 2.

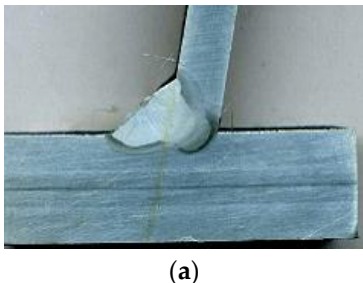
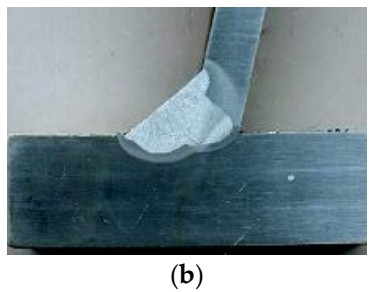
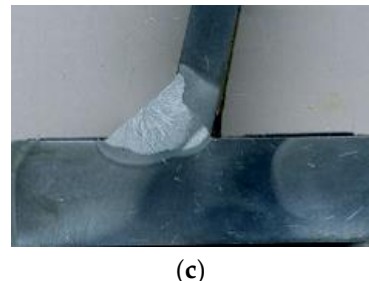

(**a**)  (**b**)  (**c**)

**Figure 2.** Macro-section of rib-to-deck single-sided welded joints; (**a**) unfused depth of 1.38 mm at the welding roots; (**b**) unfused depth of 1.32 mm at the welding roots; (**c**) unfused depth of 0.81 mm at the welding roots.

An unfused depth of about 1 mm remains at the welding roots of the rib-to-deck single-sided welded joints. The welding voltage and current must be strictly controlled during the welding process. The penetration depth of weld seams will not meet the requirement if the welding voltage and current are relatively low. If the welding voltage and current are relatively high, full penetration welding tends to occur at the weld seams, thus forming weld beads on the inner side of the longitudinal rib. Under loading, there is a prominent stress concentration effect at the welding roots. Fatigue cracks tend to emerge and develop there. A more severe initial manufacturing defect will lead to lower fatigue strength.

### 2.2. Rib-to-Deck Double-Sided Welded Joint

With specialized automated welding equipment, the double-sided welding technology of rib-to-deck welded joints of orthotropic steel decks involves welding a fillet weld seam at the inner side of the rib-to-deck welded joints, thus transferring the welding morphology of the structure from a single-sided fillet weld to double-sided fillet weld, as shown in Figure 3. Based on the differences in the penetration rate at the welding seams, we further divide the double-sided fillet weld type into the two types of a double-sided partially penetrated fillet weld and a double-sided fully penetrated fillet weld. Figure 4 shows the macro-section of the double-sided partially penetrated weld seam. Figure 5 shows the macro-section of the double-sided fully penetrated weld seam.

When the penetration rate of the double-sided partially penetrated rib-to-deck welded joints is 60%, their unfused length will be more than 2 mm. Meanwhile, when the penetration rate is above 80%, their unfused length will almost disappear, making their cross-section morphology the same as the morphology of the double-sided fully penetrated rib-to-deck welded joints. When the sizes of the weld seams satisfy the appropriate penetration depth, a closed rigid area forms at the unmelted zone of the double-sided welded joints' welding roots. This rigid area eliminates the repeated open–close phenomenon of the "crack-like" structure at the welding roots when the joints are under loading. Now, the welding roots are no longer the determining part of the welded joints' fatigue performance. The fatigue strength of the inner or outer sides of weld toes will directly determine the fatigue life of the welded joints.

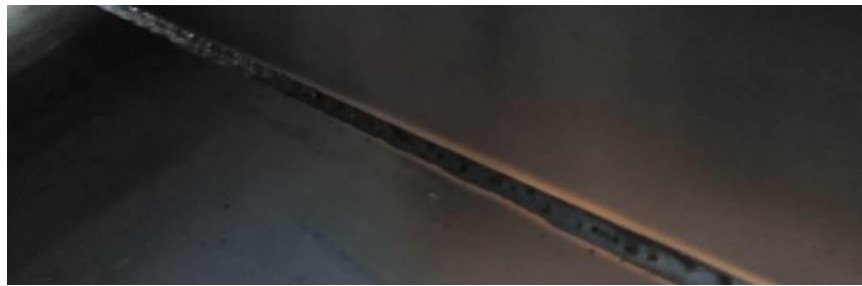

**Figure 3.** Weld seam at the inner side of rib-to-deck welded joints.

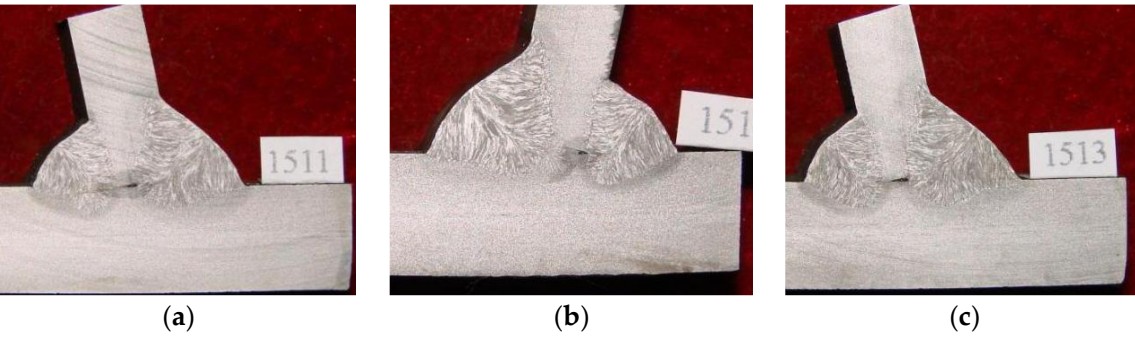

|  |  |  |
|---|---|---|
| (**a**) | (**b**) | (**c**) |

**Figure 4.** Macro-section of double-sided partial-penetration rib-to-deck welded joints: (**a**) unfused length of 2.5 mm; (**b**) unfused length of 2.1 mm; (**c**) unfused length of 2.8 mm at the welding root.

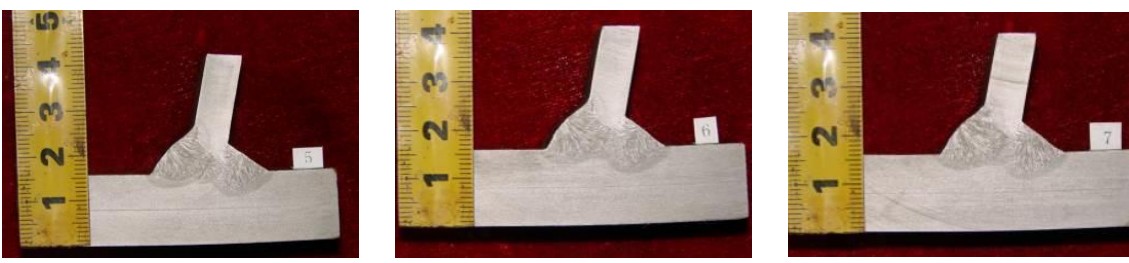

**Figure 5.** Macro-section of double-sided full-penetration rib-to-deck welded joints.

## 3. Fatigue Tests

The model test is an effective way to study the fatigue performance of rib-to-deck welded joints. Through the reasonable design of fatigue test models and loading schemes, the actual stress condition and fatigue damage accumulation process of rib-to-deck welded joints under the repeated action of vehicles can be accurately simulated. The test results can reveal the fatigue damage mechanism and the fatigue performance of welded details.

### 3.1. Specimens and Test Setup

Nine specimens were designed to compare the fatigue performances of rib-to-deck welded joints under different welding processes. These models include three single-sided welding specimens (named from S1 to S3), three double-sided partially penetrated welding specimens (named from DP1 to DP3), and three double-sided fully penetrated welding specimens (named from DF1 to DF3). The specimens have a width of 2700 mm, longitudinal lengths of 6000 mm, and a vertical height of 738 mm. Each testing model consists of three crossbeams (named 1# to 3#) and four U-ribs (named U1 to U4), with the U-ribs and deck plates extending 500 mm from the center line of the crossbeams.

The thicknesses of the plates used in the specimens are consistent with the design scheme of the Shenzhen–Zhongshan Link, as shown in Figure 6.

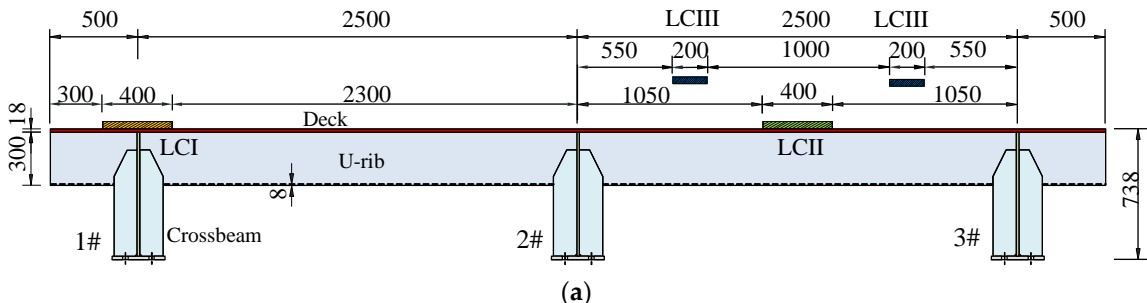

(**a**)

**Figure 6.** *Cont.*

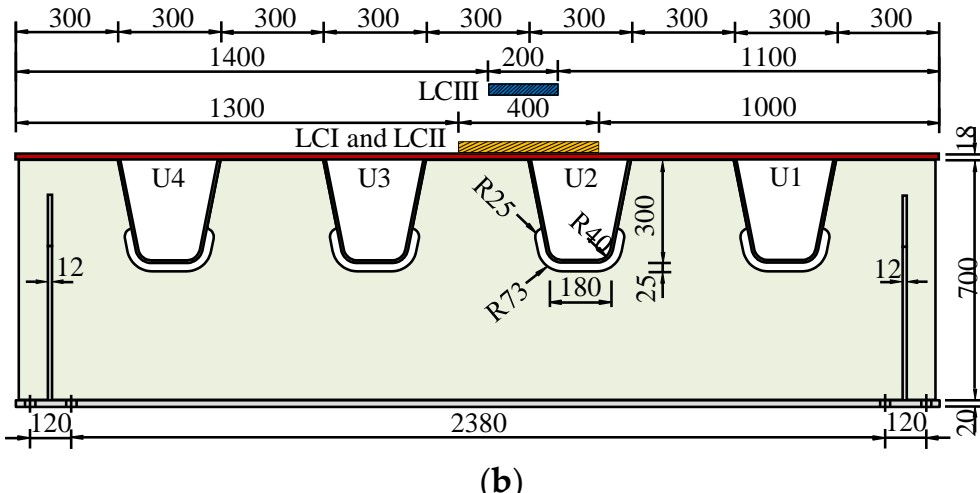

**(b)**

**Figure 6.** Specimens and loading methods: (**a**) vertical view; (**b**) side view.

Before loading the specimens, we applied ultrasonic and phased array flaw detection technologies to measure the welding seams. Then, we compared measurement results with the inspection report submitted by the production entity to verify the accuracy of the quality measuring technology of the welding seams.

### 3.2. Arrangement of Measuring Points

The selection principle of key measuring points of the fatigue specimens is as follows: (1) Measuring points should be close to the target positions of the rib-to-deck welded joints, so that the measuring results can effectively reflect the actual stress condition at the target locations. (2) It should be convenient to attach strain gauges to the measuring points selected. (3) The finite element analysis method should be applied to evaluate the possible positions of fatigue cracks. Since attaching strain gauges to the inner sides of weld toes is impossible, measuring points are arranged at the welding roots or directly above the inner sides of weld toes. Therefore, the occurrence and development of fatigue cracks can be monitored through the changes in the measuring data of strain measuring points on the upper surface. Figure 7a shows the arrangement of strain gauges on the single-sided rib-to-deck welded joints of S1 to S3. Figure 7b shows the arrangement of strain gauges on the double-sided partially penetrated rib-to-deck welded joints of DP1 to DP3. Figure 7c shows the arrangement of strain gauges on the double-sided partially penetrated rib-to-deck welded joints of DF1 to DF3. Figure 7d is a photo of the strain gauges.

In the fatigue tests, BF120-2AA type strain gauges are used, and the dynamic strain at the key measuring points is monitored in real time with a multi-channel DH5961 dynamic acquisition system. This acquisition system has the advantages of high strain measurement resolution, broad measurement range, and fast test rate. It can ensure that the system test error will not annihilate the test results, and the test requirements are satisfied. The test loading frequency is 3 Hz. The acquisition frequency of the strain data is 100 Hz, so the stress variation characteristics at the welded joints can be completely obtained. During the test, the influence of temperature changes on the test stability during the test is eliminated in real time through the temperature compensation strain gauges.

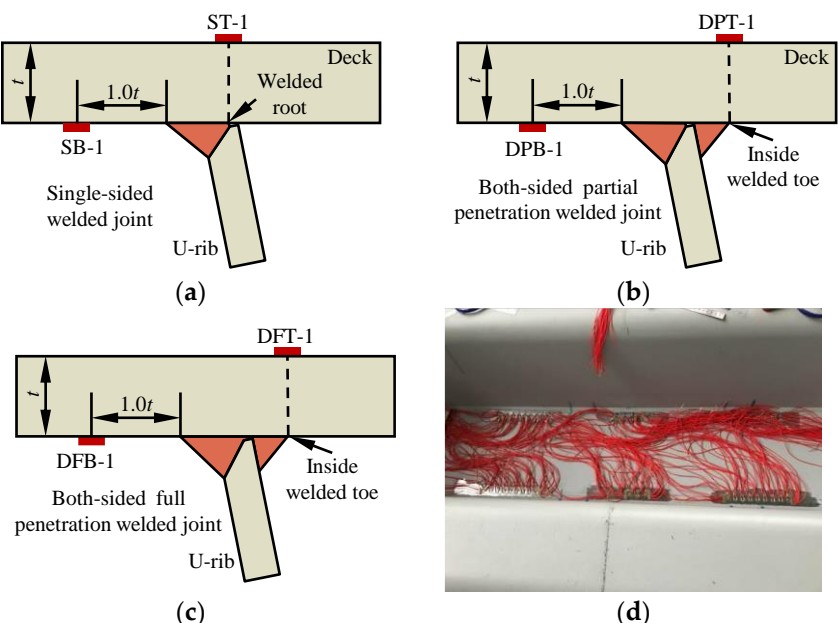

**Figure 7.** Arrangement of measuring points: (**a**) arrangement of strain gauges on the single-sided rib-to-deck welded joints.ST and SB are the strain gauges, and the number behind them indicates their location; (**b**) arrangement of strain gauges on the double-sided partially penetrated rib-to-deck welded joints. DPT and DPB are the strain gauges, and the number behind them indicates their location; (**c**) arrangement of strain gauges on the double-sided fully penetrated rib-to-deck welded joints. FPT and FPB are the strain gauges, and the number behind them indicates their location; (**d**) photo of strain gauges.

### 3.3. Loading Procedure

An MTS793 test system is used in the fatigue tests to conduct the fatigue loading. Between the loading pad and the testing model, rubber bearings are placed. To accurately simulate the actual stress condition of the critical joint structure, we apply the six-point constraint consolidation method at the lower flange of each crossbeam. The ground anchor beam at each constraint position is connected with the ground anchor through high-strength bolts. To improve the test efficiency, we use two MTS test systems to load two testing models simultaneously. Figure 8 shows the test loading.

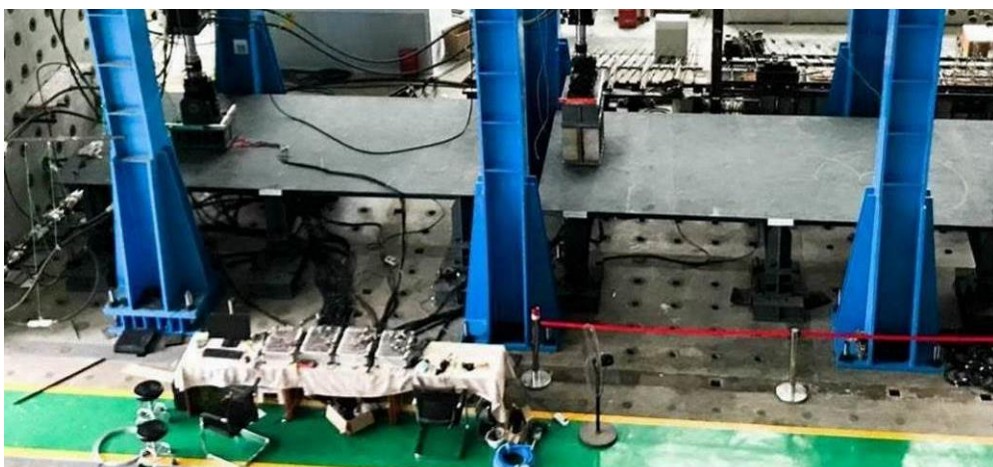

**Figure 8.** Picture of the test model loading.

To obtain the most detailed fatigue performance of the rib-to-deck welded joints possible, three loading methods shown in Figure 6 were designed. The three loading

methods (LCI, LCII, and LCIII), that are shown in Figure 6 were applied to each testing model, and the loading was conducted separately. To prevent different loading models from interfering with each other, we have applied the two loading methods on the loading one after the other. The loading positions under different loading methods are between different crossbeams of the testing model. Table 1 shows the loading details of the nine testing models.

**Table 1.** Loading procedures of the specimens.

| Specimens | Loading Method | Loading Position | Amplitude | Loading Cycles |
|---|---|---|---|---|
| S1 | LCI | Above the 1 # crossbeam | 240 kN | 2.0 million |
| | LCII | Between 2 #and 3 # crossbeam | 240 kN | 2.0 million |
| | | | 360 kN | 4.0 million |
| S2 | LCI | Above the 1 # crossbeam | 240 kN | 2.0 million |
| | LCIII | Between 2 # and 3 # crossbeam | 360 kN | 2.8 million |
| S3 | LCI | Above the 1 # crossbeam | 150 kN | 4.6 million |
| | LCII | Between 2 #and 3 # crossbeam | 300 kN | 2.0 million |
| | | | 400 kN | 1.0 million |
| | | | 450 kN | 1.6 million |
| DP1 | LCI | Above the 1 # crossbeam | 240 kN | 2.0 million |
| | LCII | Between 2 #and 3 # crossbeam | 240 kN | 2.0 million |
| | | | 360 kN | 2.0 million |
| | | | 480 kN | 1.0 million |
| DP2 | LCIII | Between 2 #and 3 # crossbeam | 360 kN | 8.0 million |
| DP3 | LCIII | Between 2 #and 3 # crossbeam | 360 kN | 8.0 million |
| DF1 | LCI | Above the 1 # crossbeam | 180 kN | 4.2 million |
| | LCII | Between 2 #and 3 # crossbeam | 300 kN | 2.0 million |
| | | | 400 kN | 1.0 million |
| | | | 450 kN | 1.2 million |
| DF2 | LCI | Above the 1 # crossbeam | 180 kN | 4.2 million |
| | LCII | Between 2 #and 3 # crossbeam | 300 kN | 2.0 million |
| | | | 400 kN | 1.0 million |
| | | | 450 kN | 2.0 million |
| DF3 | LCIII | Between 2 #and 3 #c rossbeam | 360 kN | 8.0 million |
| | | | 420 kN | 2.0 million |

## 4. Fatigue Strength Evaluation

### 4.1. Results of Fatigue Tests

After the fatigue tests, a total of 11 fatigue cracks were detected on the nine testing models. There is a significant difference in the positions of fatigue cracks under these three loading conditions. Under the LCI loading method, fatigue cracks appear directly below the load; specifically, the cracks generally appear at the intersection of the longitudinal rib, deck plate, and crossbeam and develop along the thickness direction of the deck plate (Crack-I). Under the LCII loading method, fatigue cracks generally appear at a certain distance from the loading position (Crack-II). Under the LCIII loading method, fatigue cracks often appear between two loading zones (Crack-III). Figure 9 shows all three scenarios. The test results are shown in Table 2.

**Table 2.** Information on fatigue cracks.

| Specimens | Loading Method | Crack Type | Specimens | Loading Method | Crack Type | Specimens | Loading Method | Crack Type |
|---|---|---|---|---|---|---|---|---|
| S1 | LCI | Crack-I | DP1 | LCI | Crack-I | DF1 | LCI | Crack-I |
| | LCII | Crack-II | | LCII | Crack-II | | LCII | Crack-II |
| S2 | LCI | Crack-I | DP2 | LCIII | Crack-III | DF2 | LCI | Crack-I |
| | LCIII | Crack-III | | | | | LCII | Crack-II |
| S3 | LCI | Crack-I | DP3 | LCIII | Crack-III | DF3 | LCIII | / |
| | LCII | Crack-II | | | | | | |

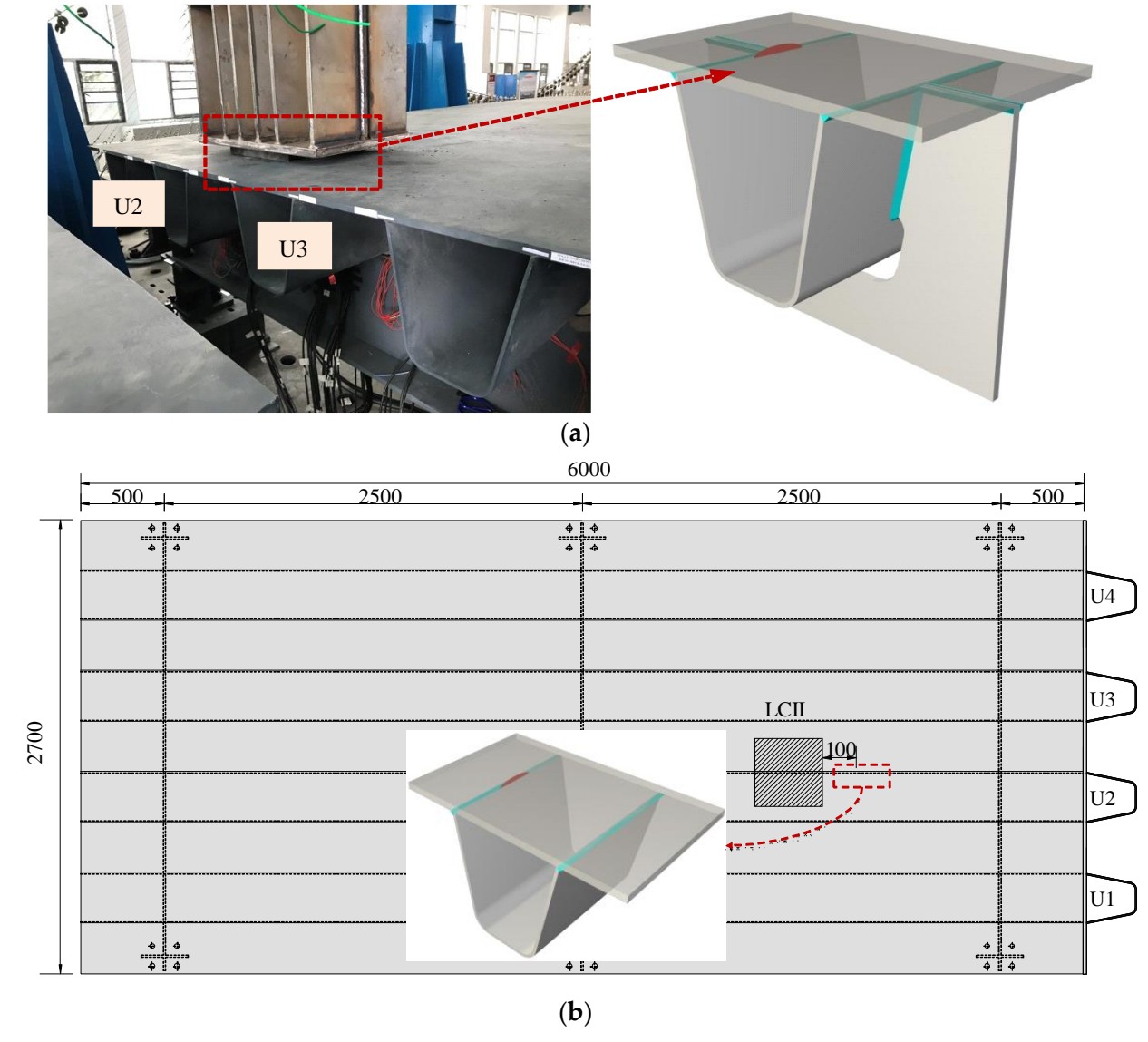

**Figure 9.** *Cont.*

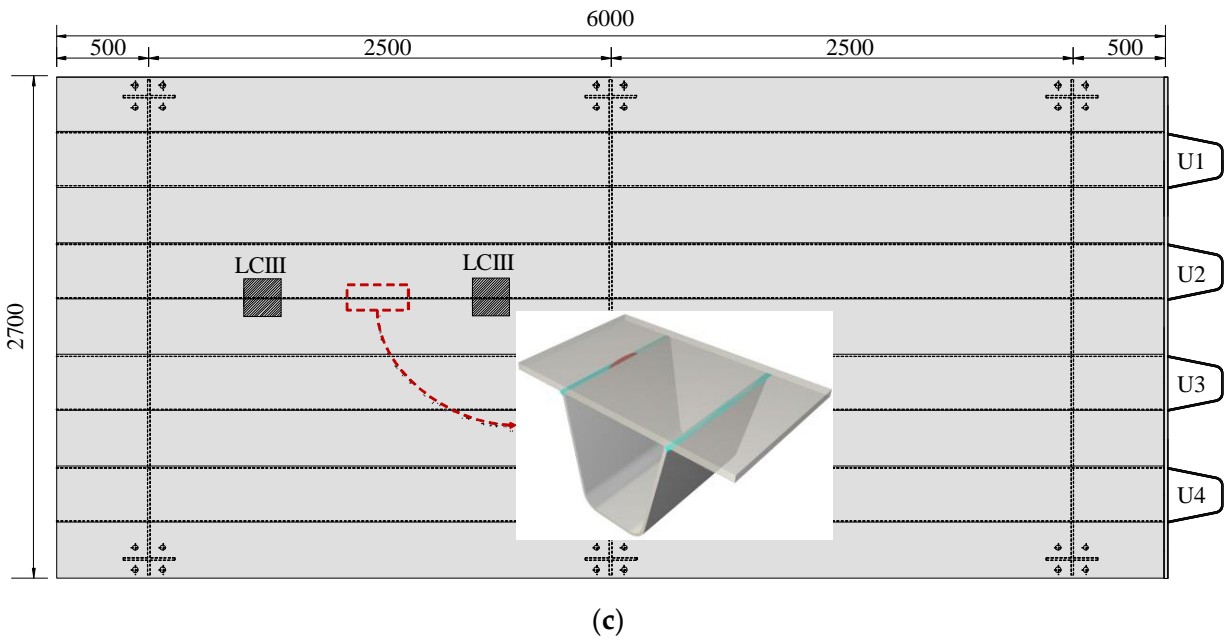

(**c**)

**Figure 9.** Typical positions of fatigue cracks: (**a**) crack-I; (**b**) crack-II; (**c**) crack-III.

The primary fatigue cracking pattern of the rib-to-deck single-sided welded joints includes the cracks occurring at the welding roots and developing along the thickness direction of the deck plate. The main fatigue cracking pattern on double-sided welding includes the cracks appearing at the inner sides of weld toes and developing along the deck plate. The typical fatigue cracks at rib-to-deck welded joints are shown in Figure 10.

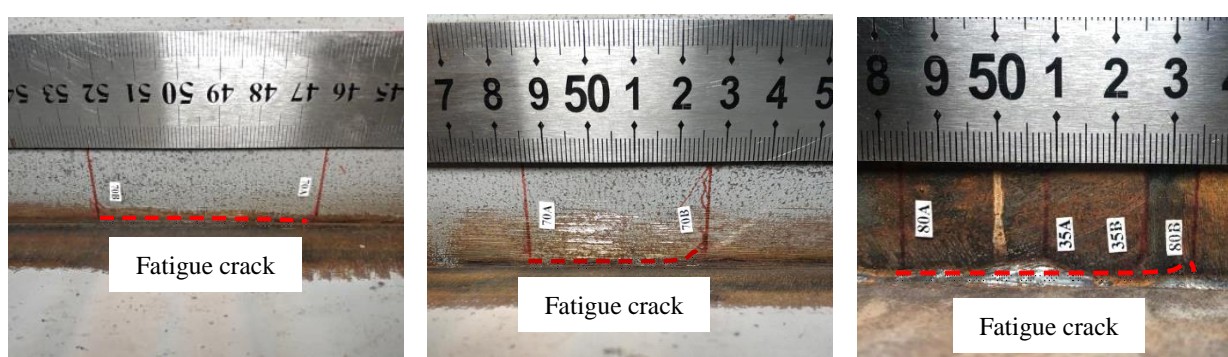

**Figure 10.** Typical fatigue cracks at rib-to-deck welded joints.

The stress-changing process of the local measuring points of the rib-to-deck welded joints of the testing model S1 under the LCII loading method (as shown in Figure 11) is taken as an example. Their local stresses will change when any fatigue crack occurs on the rib-to-deck welded joints. The reason is that the change in local stiffness caused by fatigue cracking will lead to a redistribution of local stress. Therefore, based on the stress-changing trend of the key measuring points of a testing model, the fatigue damage of rib-to-deck welded joints can be judged. After all testing models are loaded and tested, the detection results of the ultrasonic non-destructive test and the ultrasonic phased array inspection should be combined to verify the strain data. This shows that it is viable to monitor the fatigue damage status at the welding roots or welding toes of rib-to-deck welded joints by arranging measuring points on the upper surfaces of deck plates.

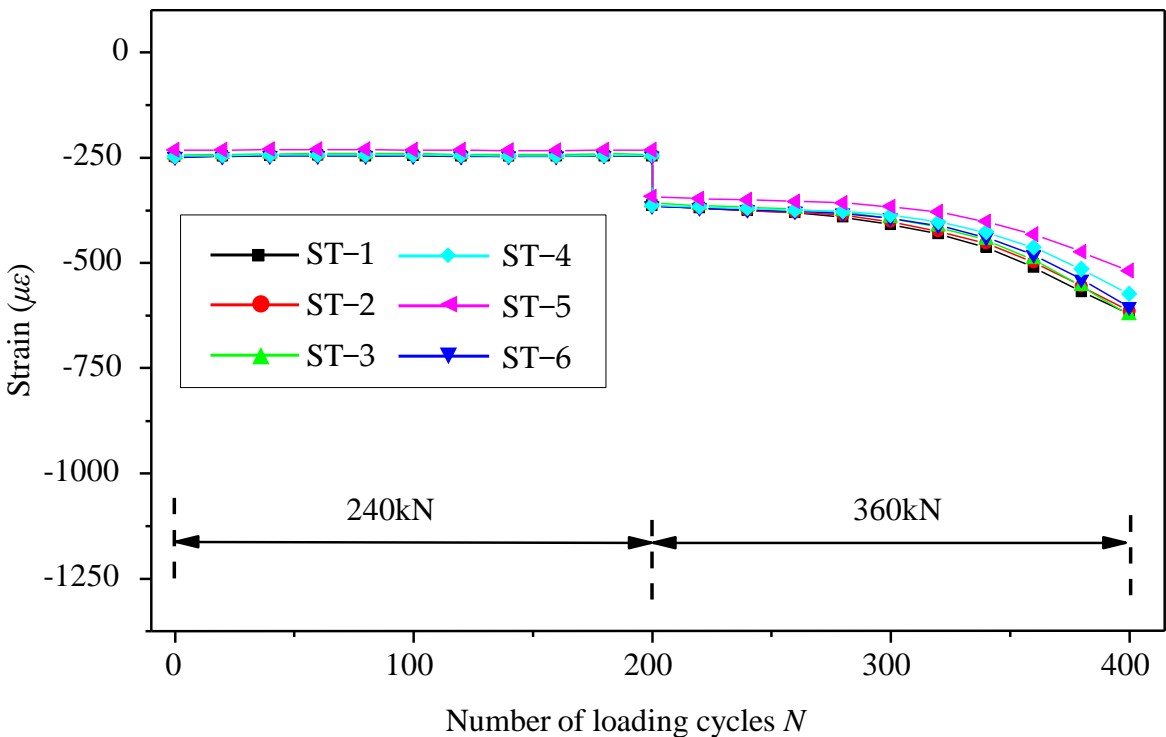

**Figure 11.** Stress-changing process.

### 4.2. Assessment of Rib-to-Deck Welded Joints

Currently, the most commonly used methods in assessing fatigue resistance mainly include the nominal stress method, the notch stress method, the structural stress method, and the fracture mechanics method. In this paper, the nominal and structural stress methods were used to evaluate the fatigue performance of the rib-to-deck welded joints.

To ease the evaluation, and regarding the two recommended fatigue failure criteria of 10% and 25% of the stress-changing amplitudes of key measuring points in accessing the fatigue strength of steel decks' vulnerable joints (Kolstein [3]), in this paper, 10% of the stress-changing amplitudes of key measuring points was used as the fatigue failure criterion.

Because it is impossible to directly measure the nominal stress and structural stress at the welding roots and toes with strain gauges, finite element models were used to measure these stresses indirectly. Figure 12 shows the finite element models, which were built by Ansys. The models use elements with eight nodes, a steel elastic modulus of 206 GPa, and a Poisson's ratio of 0.3. The parameters of plate thicknesses and weld seam sizes used in the finite element all comply with the testing models. A denser model meshing is applied in the concerned areas to improve the calculation accuracy. The meshing size and nominal stress extraction of the finite element models are consistent with reference [23], and they can avoid a change in the nominal stress caused by the meshing size. The extraction of the structural stress is consistent with that in reference [12]. The constraint effect of high-strength bolts is simulated on the test model by setting the degrees of freedom between the lower flange of the diaphragm and the joints in the anchorage area of the laboratory. The effectiveness of the finite element model is verified by comparing the measured strain data with the simulated node stress data. When the test data are in good agreement with the results of the finite element model, the finite element model is considered to represent the test model. At this time, the nominal stress and structural stress at the inner weld root or weld toe are extracted to evaluate the fatigue strength of the rib-to-deck welded joints under the cracking mode of the specimens.

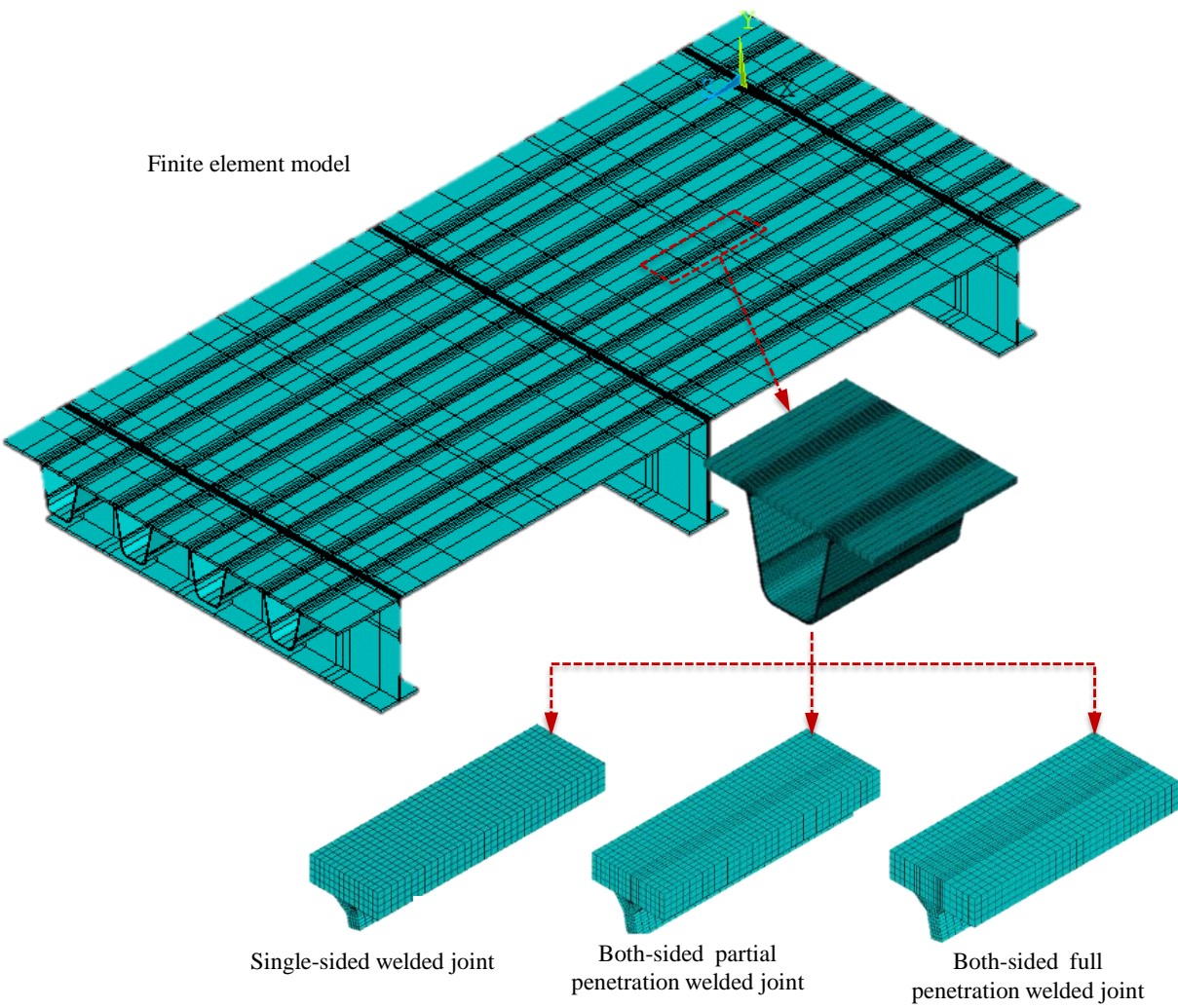

Finite element model

Single-sided welded joint

Both-sided partial penetration welded joint

Both-sided full penetration welded joint

**Figure 12.** Finite element model based on segmental specimen.

Figure 13a,b compare each testing model's fatigue strength with the nominal stress curve. Their equivalent stress amplitudes were obtained for the testing models with variable amplitude loading based on the linear damage accumulation theory.

The fatigue strength of the rib-to-deck welded joints of the testing models is higher than the Chinese standard FTA90 strength level [27], and it is also higher than the American standard fatigue strength of class C joints [28]. Compared with early manual welding, mechanical welding with higher stability is generally used at present. During the welding process, the control of heat input and welding speed is more standardized, making the quality of weld formation more stable. This may be one of the reasons for the increased fatigue strength. The fatigue strength of double-sided rib-to-deck welded joints is significantly higher than that of single-sided welded joints. It is also higher than the Chinese standard FAT112 strength level and the American standard fatigue strength of class B joints.

Meanwhile, after comparing the fatigue test results with the main *S-N* curves [12,29,30], the results show that the fatigue strength of the rib-to-deck welded joints is within a $\pm 2\sigma$ range of the main *S-N* curves (as shown in Figure 14). This indicates that compared with the nominal stress method, the divergence of the test data is lower.

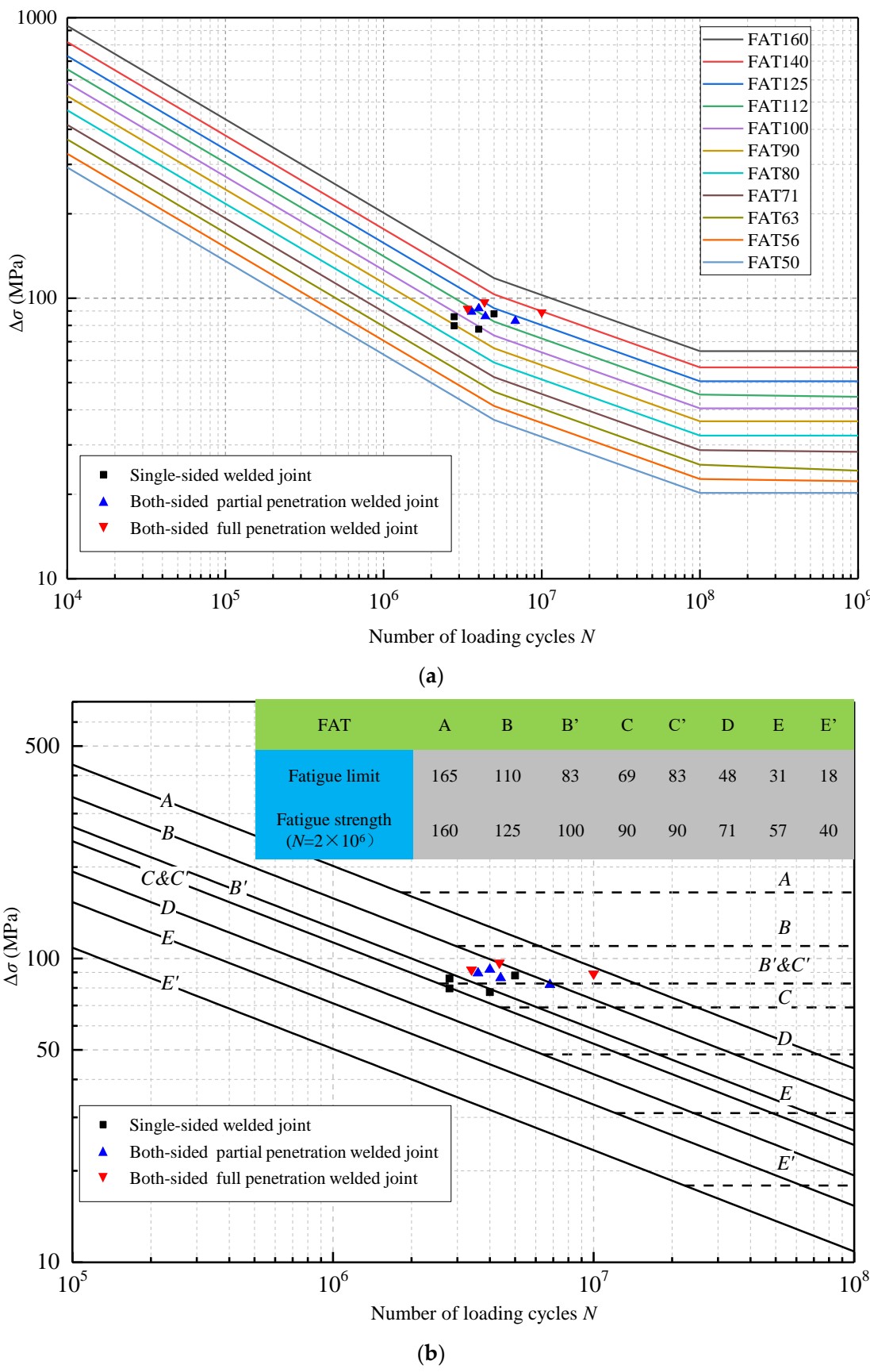

**Figure 13.** Nominal stress method based on (**a**) JTG D64-2015; (**b**) AASHTO.

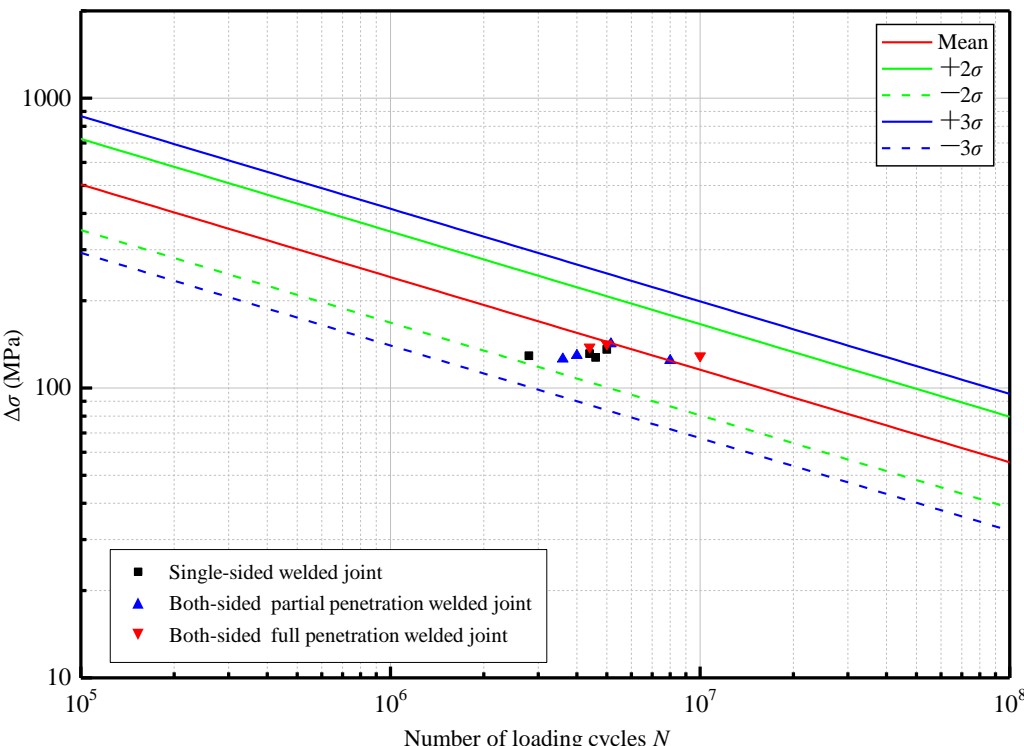

**Figure 14.** Structural stress method.

## 5. Conclusions

Double-sided full-penetration rib-to-deck welded joints do not have the same "crack-like" flaws at the welding roots as single-sided welded joints, thus reducing the probability of manufacturing defects such as weld defects during the welding process. The experimental results show that the fatigue strength of the double-sided rib-to-deck welded joints is higher than that of single-sided rib-to-deck welded joints, so the service life of orthotropic steel decks with double-sided rib-to-deck welded joints can be significantly enhanced, and their economy can be significantly improved.

1. Under the 80% penetration rate requirement of single-sided rib-to-deck welded joints, the unfused length at the welding roots is about 1 mm. Under a relatively low penetration rate, there are quite broad unfused zones between weld seams of the double-sided partially penetrated rib-to-deck welded joints. The unfused zones cause stress concentration at the welded joints, which reduces the fatigue strength of the welded joints. With a penetration rate above 80%, the forming effect of weld seams of double-sided partially penetrated welded joints is similar to the effect of double-sided full-penetration welded joints. It is suggested to adopt the double-sided welded joints in practice, and to ensure that the penetration rate is beyond 80%.

2. Fatigue cracks of specimens of single-sided welded details appear at the root of decks and develop along its thickness direction. In contrast, fatigue cracks of testing models of double-sided welded details appear at the inner-side toes of decks and develop along the thickness direction. The fatigue strength of rib-to-deck welded joints is higher than the Chinese standard of FAT90 and the American standard of FAT C joints. The fatigue strength of the double-sided rib-to-deck welded joints is significantly higher than that of single-sided welded joints.

3. The fatigue strength of rib-to-deck welded joints is within a $\pm 2\sigma$ range of the main *S-N* curves with the structural stress method. Compared with the nominal stress method, the divergence of the structural stress method is lower. It is also convenient to use the structural stress method to evaluate the fatigue performance of welded joints.

4.  To further enhance the fatigue resistance of rib-to-deck welded joints, it is important to optimize the appearance-forming quality of the inner and outer sides of the weld toes. As the structure system of orthotropic steel bridge decks is complex and contains various kinds of welded details, the overall service quality can only be improved by improving the fatigue strength of the rib-to-deck welded joints. Therefore, it is still necessary to systematically improve the service life from the perspective of the structural system.

5.  This paper verifies that the fatigue strength of double-sided rib-to-deck welded joints is higher than that of the single-sided rib-to-deck welded joints used by existing bridges. However, the loading method adopted in the test is still very different from the actual bridge load, so it fails to explore the fatigue strength of all possible failure modes of different welded joints. The trial data are limited, and further tests are needed to validate the results.

**Author Contributions:** Conceptualization, C.F.; methodology, L.D.; software, L.D.; validation, L.D.; formal analysis, L.D.; investigation, K.W.; resources, S.S.; data curation, L.D.; writing—original draft preparation, L.D.; writing—review and editing, K.W.; visualization, H.C.; supervision, S.S.; project administration, C.F.; funding acquisition, C.F. All authors have read and agreed to the published version of the manuscript.

**Funding:** Research and Development Projects in Key Areas of Guangdong Province, grant number 2019B111106002.

**Institutional Review Board Statement:** The study did not require ethical approval.

**Informed Consent Statement:** The study did not require ethical approval.

**Data Availability Statement:** Data will be made available on request.

**Acknowledgments:** We would like to thank KetengEdit (www.ketengedit.com (accessed on 10 December 2022)) for its linguistic assistance during the preparation of this manuscript.

**Conflicts of Interest:** The authors declare no conflict of interest.

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
