# Peer review of "Fatigue Tests and Failure Mechanism of Rib-to-Deck Welded Joints in Steel Bridge"

_sustainability, doi:10.3390/su15032108_

Round 1

Reviewer 1 Report

1. The results presented in the abstract should be quantitative.

2. The history of research is not well written. You should clearly summarize the work the researchers have done and their important results.

3. Provide a flowchart of the essay process at the end of Section 1.

4. Mention the standards used in conducting the tests.

Reviewer 2 Report

Dear Authors, 

The study is interesting and will make important contributions to practice. Although the subject of the study was chosen quite well, this was not revealed in the study. Suggested corrections;

1. The literature section is very poor. Detailed literature information should be given in the study. Please check these and similar references;

Cheng et al., Experimental study on fatigue failure of rib-to-deck welded connections in orthotropic steel bridge decks

Yang Liu et al Fatigue performance of rib-to-deck double-side welded joints in orthotropic steel decks

Jun lie t al. An equivalent structural stress-based fatigue evaluation framework for rib-to-deck welded joints in orthotropic steel deck

Mairone et al Fatigue Performance Analysis of an Existing Orthotropic Steel Deck (OSD) Bridge

2. Please add the stages of your study to the introduction with a paragraph.

3. Explain more clearly the difference and novelty of your work from other works.

4. Please correct the title of Table 3.

5. In the abstract section, please briefly add the numerical results you have obtained.

6. The conclusion part should definitely be expanded.

7. Demonstrate the applicability of your work, especially in practice.

8. Please include the contribution of your work to similar studies in the future.

9. Include the limitations of your work in the conclusion.

10. If possible, I recommend comparing your results with results from other studies.

 Yours Sincerely

Reviewer 3 Report

The author(s) have concluded that the fatigue strength of rib-to-deck welded joints of the testing models of the Shenzhen - Zhongshan Link Project, is higher than the Chinese standard strength level, and the American standard fatigue strength of Class C joints without explaining the scientific reason behind that.  

Author Response

Reviewer #3:

The author(s) have concluded that the fatigue strength of rib-to-deck welded joints of the testing models of the Shenzhen-Zhongshan Link Project, is higher than the Chinese standard strength level, and the American standard fatigue strength of Class C joints without explaining the scientific reason behind that.

Reply:

The authors would like to sincerely thank the editor and reviewers for their time and efforts to review the revised manuscript. The following presents our replies to the specific review comments marked in blue color. Revised portions are marked in red in the revised manuscript.

Many thanks for the comments and suggestions. Based on the analysis of the existing research results of the author's team, the reason why the fatigue strength of the test model described in this paper is generally higher than the Chinese and American standards may be that the new welding process has smaller-scale manufacturing defects. Compared with the early manual welding, mechanical welding with higher stability is generally used at present. During the welding process, the control of heat input and welding speed is more standardized, making the quality of weld formation more stable. Revised portions are marked in red in the revised manuscript.

Reviewer 4 Report

Fatigue Tests and Failure Mechanism and of Rib-to-Deck Welded Joints in Steel Bridge   

Authors:   Chuan-bin Fan, Le-tian Da , Kang-chen Wang, Shen-you Song and Huan-yong Chen

Summary:

In this study, the authors studied the failure mechanisms due to fatigue in rib to deck welded joints primarily used in Steel Bridges. In this work, the authors compared the morphological differences of rib-to-deck welded joints for various welding processes such as single and double weld joints. They have conducted fatigue tests on nine groups of testing models. Additionally, with the help of nominal stress method and the structural stress method, they have analyzed and compared the fatigue performance of the rib-to-deck welded joints of single sided welding, double-sided partially penetrated welding, and double-sided fully penetrated welding. Their study concludes that the double-sided rib-to-deck welded joints do not have the “crack-like” structure at  the welding roots as the single-sided welded joints and reduce the probability of manufacturing defects like weld beads during the welding process. Their study also reveals that the fatigue cracking of the double-sided rib-to-deck welded joints is now at the inner sides of weld toes but not at the welding roots as the single-sided welded joints, thus significantly improving their fatigue strength. The authors are also included sufficient number of references in the manuscript.

Recommendation: 

After reviewing the paper, it is recommended that this manuscript shall not be acceptable for publication in MDPI-Sustainability Journal without undergoing major revision. Some of the corrections identified are given below

Comments to Authors:

1.       The usage of grammar and English are generally good. However, in some places spelling,  and grammatical mistakes and incoherent sentence are noticed.  Hence, it is recommended to carry out a detailed proof reading of the manuscript that would improve the readability of the paper further.

2.       The abstract not capturing the actual work done. It is recommended to rewrite for clarity.

3.       In page 2, line 39, the words “Table 3. [9-11]” is not conveying any meaning.

4.       It is suggested to expand the abbreviation “UHPC” in line 46. It is also suggested to expand the abbreviations when they first appear first time in other places if applicable.

5.       It is suggested to explain in detail of various abbreviations used in related to figure 6, 7 and table 1. Particularly, it is not clear LCI, LCII, U1, U2, U3 , U4, SB1, ST1, DPT, DPB etc.

6.       In figure 7, sub figure (d) is not explained in captions. It is also suggested to rewrite the corresponding description of the figure in related to sub figures instead of single figure for clear understanding.

7.       In Table 1, the loading cycle values provided are actual test data?

8.       It is suggested to explain in detail about various types of crack appear in the specimen (Section 4).

9.       The main issue of this manuscript is related to FEA analysis presented in section 4.2. The authors not provided how the fatigue analysis carried out, what type of software is used, what is the role of mesh count on the results (if any mesh independence study is carried), initial and boundary conditions used, various steps involved to finding the results etc.

10.   It is recommended to include steps or flow chart illustrate the FEA analysis used in this work.

11.   It is not clear if  the data shown in Figure 13 and Figure 14 came from FEA analysis or experiment. 

Round 2

Reviewer 1 Report

.

Reviewer 4 Report

--NA---